# Mechanism of TCF21 Downregulation Leading to Immunosuppression of Tumor-Associated Macrophages in Non-Small Cell Lung Cancer

**DOI:** 10.3390/pharmaceutics15092295

**Published:** 2023-09-07

**Authors:** Hong Liu, Run He, Xuliang Yang, Bo Huang, Hongxiang Liu

**Affiliations:** 1Department of Thyroid Oncology, Chongqing University Cancer Hospital, Chongqing 400030, China; doctorlh2015@163.com; 2School of Biological and Chemical Engineering, Chongqing University of Education, Chongqing 400067, China; 13594300654@139.com; 3Department of Thoracic Surgery, Chongqing Hospital of Traditional Chinese Medicine, Chongqing 400011, China; yangxuliang1027@163.com (X.Y.); 18983765290@163.com (B.H.)

**Keywords:** non-small cell lung cancer, TCF21, Notch4, macrophage, immunoregulation

## Abstract

Lung cancer, as one of the high-mortality cancers, seriously affects the normal life of people. Non-small cell lung cancer (NSCLC) accounts for a high proportion of the overall incidence of lung cancer, and identifying therapeutic targets of NSCLC is of vital significance. This study attempted to elucidate the regulatory mechanism of transcription factor 21 (TCF21) on the immunosuppressive effect of tumor-associated macrophages (TAM) in NSCLC. The experimental results revealed that the expression of TCF21 was decreased in lung cancer cells and TAM. Macrophage polarization affected T cell viability and tumor-killing greatly, and M2-type polarization reduced the viability and tumor-killing of CD8^+^T cells. Meanwhile, overexpression of TCF21 promoted the polarization of TAM to M1 macrophages and the enhancement of macrophages to the viability of T cells. Furthermore, there appears to be a targeting relationship between TCF21 and Notch, suggesting that TCF21 exerts its influence via the Notch signaling pathway. This study demonstrated the polarization regulation of TAM to regulate the immunosuppressive effect, which provides novel targets for the treatment of lung cancer.

## 1. Introduction

Non-small cell lung cancer (NSCLC) is the most common histological subtype of lung cancer, accounting for 85% of the overall incidence of lung cancer, and it ranks first among male malignant tumors in China [1,2]. As the cancer cells of NSCLC spread slowly, and the early onset of patients lacks typical symptoms, a majority of patients are diagnosed at an advanced stage and have missed the best treatment period for the tumor, resulting in a shortened period of survival [3]. Despite advancements that have been achieved in clinical and experimental oncology in recent years, the 5-year survival rate of NSCLC is still at a low level [4]. Thus, exploring the molecular mechanism of NSCLC and identifying molecular markers related to NSCLC progression would eventually produce remarkable clinical significance in improving the prognosis of patients.

Certain recent studies have indicated the importance of tumor-associated macrophages (TAM) in the TME, NSCLC tumor progression, angiogenesis, and distant metastasis [5], and they serve as an important factor in the prognosis of patients. It is hoped that through the in-depth study of TAMs, the microenvironment of the tumor matrix can be modified and reshaped to improve the therapeutic effect of drugs. TAMs act in concert with tumor cells to promote tumor invasion and metastasis [6]. Taken together, the exploration of the regulation of the TAM phenotype and its action mechanism on the development of lung adenocarcinoma is helpful for identifying new targets and novel drugs for the treatment of lung cancer.

Based on a bioinformatics analysis approach, this study screened out transcription factor 21 (TCF21), a markedly down-regulated gene in NSCLC. TCF21 is a recently discovered tumor suppressor gene characterized by reversing epithelial–mesenchymal transition (EMT), and it is of great significance to the growth and differentiation of cells [7]. The role of TCF21 in NSCLC is unclear, and we hypothesize that TCF21 may contribute to the immunosuppressive effects of TAMs in this context. Therefore, this study sought to elucidate the mechanism by which TCF21 downregulation induces immunosuppression in NSCLC-associated TAMs, with the goal of uncovering potential targets for NSCLC immunotherapy.

## 2. Materials and Methods

### 2.1. Differentially Expressed Gene (DEG) Analysis

Relevant NSCLC data were downloaded from the TCGA database with the dataset of TCGA-LUAD. The R package DEseq2 was applied to analyze the DEGs of the samples. After the *p*-value was calculated, multiple hypothesis testing was used for correction. The threshold of the *p*-value was determined by controlling the false discovery rate (FDR), and the corrected *p*-value was used as the q-value. The differential expression fold was subsequently calculated based on the FPKM value and expressed as Fold-change. The screening indicators for this analysis were *p*-value < 0.05, log2FC > 1 or <−1. Enrichment analysis of the Gene Ontology (GO) and Kyoto Encyclopedia of Genes and Genomes (KEGG) pathways for DEGs was performed using the clusterProfiler (v4.2.2) package.

### 2.2. Cell-Grouping Treatment

Human lung cancer cells (A549) were purchased from the BeNa Culture Collection. Healthy human lung epithelial cells (BEAS-2B) and human mononuclear cells (THP-1) were purchased from Procell Life Science & Technology Co., Ltd. (Wuhan, China). The density of THP-1 cells was adjusted to 2 × 10^5^ cells/mL and added to 6-well plates for culture at 37 °C in a 5% CO_2_ cell incubator. The THP-1 cells were randomly divided into two groups, namely the M group and TAM group, and were treated as follows: Group M: Cells were induced and cultured in a RPMI-1640 medium containing 50 ng/mL PMA for 5 d [8]; TAM group: The A549 cells, maintained in a healthy growth state, were harvested with the same passage method. The cell density of both A549 cells and PMA-induced macrophages was adjusted to 1 × 10^5^ cells/mL and co-cultured in a Transwell system. A549 cells were seeded in the upper chamber, while PMA-induced macrophages were placed in the lower chamber. They were co-cultured in equal proportions for 48 h.

### 2.3. Induction and Grouping of M1 and M2 Macrophages

Well-grown THP-1 cells were collected into a centrifuge tube and centrifuged at 1200 rpm for 3 min. After the supernatant was discarded, the cells were supplied with an RPMI-1640 medium and aspirated gently until the formation of cell suspension. The obtained cell suspension was supplemented with an RPMI-1640 medium containing 50 ng/mL PMA and mixed well. A 10 μL cell suspension was taken and mixed with 10 μL of 0.4% trypan blue staining solution and counted using a hemocytometer. The cell density was set to 2 × 10^5^ cells/mL, added into 6-well plates, and cultured in a 5% CO_2_ incubator at 37 °C for 5 d. The cells were randomly divided into three categories: the M1 macrophage group, the M2 macrophage group, and the TAM group, and followed by the following treatment: the M1 macrophage group: The cells were cultured for 24 h using a complete medium containing 200 ng/mL LPS + 20 ng/mL IFN-γ. The M2 macrophage group: The cells were cultured for 24 h using a complete medium containing 20 ng/mL IL-4 + 20 ng/mL IL-10. TAM group: The A549 cells in a good growth state were digested and harvested using the same method as passage, the density of A549 cells and macrophages induced by PMA were adjusted, and the A549 cells were inoculated in the upper chamber of the Transwell, and the macrophages induced by PMA were inoculated in the lower chamber of the Transwell and co-cultured in equal proportions for 48 h. Note: The TAM group was co-cultured for 24 h before being treated with the M1 and M2 macrophage groups [9,10].

### 2.4. Experimental Cell Grouping on Macrophage Polarization of T Cell Viability and Tumor Killing

CD8^+^T cells were isolated and purified using MACSxpress Whole Blood CD8 T cell Isolation Kit (130-098-194, NovoBiotechnology Co., Ltd., Beijing, China) and identified through flow cytometry. CD8^+^T cell group: CD8^+^T cells were resuspended in a complete medium, seeded in 6-well plates, and cultured for 4 d. M1 macrophage + CD8^+^T cell group: CD8^+^T cells were resuspended in a complete medium and inoculated in the upper chamber of the Transwell, M1 macrophages were inoculated in the lower chamber of the Transwell, CD8^+^T cells and M1 macrophages were co-cultured in an equal proportion for 4 d. M2 macrophage + CD8^+^T cell group: CD8^+^T cells were resuspended in a complete medium and inoculated in the upper chamber of the Transwell, M2 macrophages were inoculated in the lower chamber, and CD8^+^T cells and M2 macrophages were co-cultured in an equal proportion for 4 d. TAM + CD8^+^T cell group: CD8^+^T cells were resuspended in a complete medium and inoculated in the upper chamber of the Transwell, TAM was inoculated in the lower chamber, and CD8^+^T cells and TAM macrophages were inoculated in an equal proportion for 4 d.

### 2.5. qPCR Assay

The tissue was lysed with the RNAiso Plus lysate, and total RNA was subsequently extracted; the kit Goldenstar™ RT6 cDNA Synthesis Kit Ver.2 was applied for reverse transcription, and the fluorescence quantification was performed as per instructions of the kit 2 × T5 Fast qPCR Mix (SYBR Green I). Primer sequences were listed in Table 1.

### 2.6. Western Blot Assay

The tissue was lysed with RIPA lysate, and the total protein was extracted; the bands were separated using electrophoresis, and the membrane was transferred at a constant current of 250 mA. A primary antibody was diluted with primary antibody diluent at 1:1000 and incubated overnight at 4 °C. A secondary antibody was diluted to a certain concentration (1:2000) with a blocking buffer and incubated for 1 h. The ECL exposure solution was mixed with liquid A and B at a 1:1 rate and then evenly covered on the entire membrane. After the reaction for 1 min, it was loaded in the exposure meter for detection. The Western blot bands were quantified via Image-J software (National Institutes of Health, Bethesda, MD, UAS). The primary antibodies used were as follows: TCF21 (abclonal, A17451), Notch4 (abclonal, A8303), Hes1 (abclonal, A0925), Hey1 (abclonal, A16110), and GAPDH (abclonal, A19056), with the secondary antibody provided by abclonal (AS014).

### 2.7. Flow Cytometry

The cells, after centrifugation, were gently aspirated, supplied with 100 μL of special fixative A (FIX & PERM), mixed well, and incubated at room temperature for 15 min in the dark. A total of 1 mL PBS was added to the suspension after incubation, mixed well, and centrifuged at 1200 rpm for 5 min. The supernatant was discarded, with the operation repeated once. The cells, after centrifugation, were gently aspirated, added with 100 μL of special membrane-breaking agent B (FIX & PERM), mixed well, and incubated at room temperature in the dark for 20 min. After being added with 5 μL of corresponding antibodies to each tube, the cells were vibrated and mixed gently, and the specific cytokines in the cells were stained and incubated for 20 min at room temperature in the dark. A total of 1 mL PBS was added to the suspension after incubation, mixed well, and centrifuged at 1200 rpm for 5 min. The supernatant was removed, and this step was repeated once. After that, 100 μL of PBS was added to resuspend the cells, which were then subjected to flow cytometry for analysis. The percentage of apoptotic cells was determined by CytoFLEX flow cytometry (Beckman Coulter, Brea, CA, USA) and FlowJo V10 software. FIX&PERM Kit (MultiSciences (LiankeBio), Hangzhou, China, GAS003/2, A10241); Anti-Human CD86 (Biolegend, San Diego, CA, USA, 374204, B270127); Anti-Human CD206 (BD, USA, 551135, 38855); Anti-Human CD163 (BD, Franklin Lakes, NJ, USA, 563697, 57582); and Anti-Human HLADR (BD, Franklin Lakes, NJ, USA, 555560, 37681).

### 2.8. Immunofluorescence Assay

The cell slides were fixed first, added with an appropriate amount of 0.3% Triton X-100 permeabilization solution, and incubated at room temperature for 5 min. The cells were blocked with goat serum at room temperature for 30 min. After removing and drying the blocking solution, they were incubated with a primary antibody (diluted at a ratio of 1:500) and a secondary antibody (diluted at a ratio of 1:50). DAPI was added by drops for incubation in the dark, and the specimens were then nucleated. The slides were sealed with an anti-fluorescence quencher. The sections were photographed via an inverted Mshot MF53 microscope produced by Guangzhou Micro-shot Technology Co., Ltd. Ki67 Rabbit pAb (ABclonal, A11390) and FITC Goat Anti-Rabbit IgG (ABclonal, AS011).

### 2.9. ELISA Assay

ELISA experiments were performed with the Human IFN-γ ELISA Kit (Ruixin Biotech, China, RX106205H, 202208). We first prepared the well plates required, set standard wells and sample wells, and added 50 μL of standard substances at different concentrations to each standard well; then, the sample wells were supplemented with 50 μL of samples to be tested; and finally, detection antibody labeled by horseradish catalase 100 μL was supplemented to each standard solution well and sample well respectively. Following the plate was blocked with a sealing membrane, it was incubated at 37 °C for 60 min. After the incubation was completed, the sealed membrane was uncovered, and the liquid of the wells was abandoned. The samples were dried with absorbent paper; each well was filled with a washing solution and left to stand for 20 min. Afterward, the washing solution was then discarded, and the samples were dried again on absorbent paper. The procedures were repeated five times. Substrate A and B were mixed at a ratio of 1:1, and 100 μL was added to each well, which was then sealed with a piece of sealing film and incubated at 37 °C in the dark for 15 min. When the incubation was completed, termination solution at a quantity of 50 μL was added to each well. The OD value of each well was measured at 450 nm wavelength within 15 min.

### 2.10. Dual-Luciferase Reporter Assay

The binding site between TCF21 and the gene promoter Notch was predicted using an analysis website. ThebTCF21 overexpression vector was synthesized and constructed. Meanwhile, the Notch gene promoter sequence 2000 bp (WT) was designed and synthesized as well as the Notch sequence mutated at this site 2000 bp (mut). The corresponding luciferase reporter gene vector was subsequently constructed, respectively. Plasmid co-transfection was carried out with a transfection kit (Biomedicine, Shanghai, China, 10668-006), and a luciferase detection kit (Promega, Madison, WI, USA) was used for detection.

### 2.11. Chromatin Immunoprecipitation-qPCR Detection (ChIP-qPCR)

After the tissue samples were cross-linked via a vertical mixer, the nuclei were prepared using a Protease Inhibitor and DTT. A non-contact automatic ultrasonic crusher was applied to crush the cells. The ultrasonically lysed samples were added with 75 μL of agarose beads, incubated on a vertical mixer at 4 °C for pre-binding for 60 min, and centrifuged at 4000× *g* at 49 °C for 1 min. The supernatant sample was divided into two as per 0.8 mL IP and 0.1 mL Input. The Input group was supplied with 300 μL 1 × Elution Buffer and preserved at −20 °C; the IP group was supplied with ChIP antibody of IgG antibody, vertically mixed well and incubated at 4 °C for 16 h (overnight); and the IP group was added with 20 μL of prepared protein A/G-beads, mixed well by inverting on a vertical mixer, and incubated at room temperature for 30 min. The magnetic beads were washed with a wash buffer and collected with a magnetic stand, and the supernatant was then removed. Subsequently, the IP samples were eluted with an elution buffer. The samples of both the Input group and IP group were placed in a warm bath at 65 °C for 6 h for reverse cross-linking. Each tube was added with 200 μL 1 × TE and 8 μL RNaseA, inverted and mixed well 10–15 times, and incubated at 37 °C for 0.5–2 h. Each tube was then added with 4 μL EDTA and 4 μL Proteinase K, bathed at 55 °C for 2 h, followed by DNA precipitation and detected enriched DNA. A total of 20 μL of DNA samples was applied to detect DNA sequence through high-throughput sequencing technology.

## 3. Results

### 3.1. Screening and Validation of Key Genes for NSCLC Treatment Using a Bioinformatics Approach

A total of 5444 DEGs positively associated with NSCLC were screened out through bioinformatics analysis, of which 2059 were down-regulated, and 3385 were up-regulated (Figure 1A). The mostly enriched GO pathways included channel activity, signaling receptor activator activity, and passive transmembrane transporter activity, and the KEGG signaling pathways included neuroactive ligand–receptor interaction, cytokine–cytokine receptor interaction, neutrophil extracellular trap formation, calcium signaling pathway and cAMP pathway (Figure 1B,C). qPCR and WB detected the expression of TCF21 and Notch4 mRNAs and proteins in the cells of each group, implying that as compared with the normal cell BEAS-2B, the expression of TCF21 in lung cancer cell A549 was substantially increased, while Notch4 was markedly decreased; moreover, the expression of both TCF21 and Notch4 were markedly decreased in the TAM group compared with the M group (Figure 2). Therefore, we chose to verify the regulatory effect of TCF21 on NSCLC in TAM.

### 3.2. The Expression of TCF21 in Different Types of Macrophages

The expression of TCF21, Notch4, Hes1, and Hey1 in macrophages was detected via qPCR and WB assays, indicating that the expression of the previously described factors in M2 and TAM groups was markedly reduced as compared with the M1 group (Figure 3A,B). The results of qPCR detection showed that the levels of TNF-α, IL-1β, and p40 mRNAs in the M2 and TAM groups were greatly lower than those in the M1 group, while the levels of MRC1, FN1, and COX2 mRNAs were markedly higher than those in the M1 group (Figure 3C). The expression of CD86, HLADR, CD163, and CD206 in macrophages was detected via flow cytometry, and the results showed that compared with the M1 macrophage group, the positive expression percentages of CD86 and HLADR in the M2 macrophage group and TAM group decreased with a significant difference; while the positive expression percentage of CD163 and CD206 increased with a significant difference (Figure 4).

### 3.3. Impacts of Macrophage Polarization on T Cell Viability and Tumor Killing

The level of Ki67 in T cells was detected by immunofluorescence (Figure 5A), which revealed no significant difference in the M1 + CD8^+^T cell group compared with the CD8^+^T cell group; expression of Ki67 in the M2 + CD8^+^T cell group and TAM + CD8^+^T cell group was greatly decreased, implying that TAMs polarized to M2 type and T cell viability was weakened. Results of ELISA indicated no significant difference in the content of cytokine IFN-γ in the M1 + CD8^+^T cell group as compared with the CD8^+^T cell group; content of cytokine IFN-γ in M2 + CD8^+^T cell group and TAM + CD8^+^T cell group decreased, and there were significant differences (Figure 5D).

Apoptosis detection of tumor cells using flow cytometry (Figure 5B). Compared with the CD8^+^T cell group, the absolute percentage of trace-labeled tumor cells in the M1 + CD8^+^T cell group had no significant difference, indicating that compared with the CD8^+^T cell group, M1 + CD8^+^T cell group exerted little influence on tumor cell killing. The absolute percentage of trace-labeled tumor cells in the M2 + CD8^+^T cell group and TAM + CD8^+^T cell group increased, indicating that compared with the CD8^+^T cell group, M2 + CD8^+^T cell group, and TAM + CD8^+^T cell group was weakened in killing tumor cells and produced a significant effect (Figure 5C).

### 3.4. Impacts of TCF21 in Macrophages on T Cell Viability and Tumor Killing

The results of qPCR and WB detection showed that the expression of TCF21, Notch4, Hes1, and Hey1 were markedly decreased in M2 + CD8^+^T cell group and TAM + CD8^+^T cell group as compared to M1 + CD8^+^T cell group; expression of the described cytokines in TAM + CD8^+^T cell group was greatly lower than that in over-expression (OE)-TCF21 + TAM + CD8^+^T cell group (Figure 6A,B). Results of the qPCR assay confirmed that levels of TNF-α, IL-1β, and p40 mRNAs in the M2 + CD8^+^T cell group and TAM + CD8^+^T cell group were greatly lower than those in the M1 + CD8^+^T cell group; whereas those of OE-TCF21 + TAM + CD8^+^T cell group were substantially higher than TAM + CD8^+^T cell group. Compared with the M1 + CD8^+^T cell group, expression of MRC1, FN1, and COX2 mRNAs in M2 + CD8^+^T cell and TAM + CD8^+^T cell groups were markedly increased, whereas those of OE-TCF21 + TAM + CD8^+^T cell group were markedly lower than TAM + CD8^+^T group cells (Figure 6C).

The results of flow cytometry showed that, compared with M1 + CD8^+^T cell group, the positive expression percentage of CD86 in M2 + CD8^+^T cell group and TAM + CD8^+^T cell group decreased; while that of OE-TCF21 + TAM + CD8^+^T cell group was markedly higher than TAM + CD8^+^T cell group; further, there was a significant difference in the percentage decrease of HLADR positive expression among the M2 + CD8^+^T cell group, TAM + CD8^+^T cell group, and OE-TCF21 + CD8^+^T cell group; and there was a significant difference in the percentage increase of CD163 and CD206 positive expression in M2 + CD8^+^T cell group and the TAM + CD8^+^T cell group, whereas there was no significant difference of the two factors in the OE-TCF21 + CD8^+^T cell group. Compared with the TAM + CD8^+^T cell group, the positive expression percentages of CD163 and CD206 in the OE-TCF21 + CD8^+^T cell group were markedly decreased, while those of CD86 and HLADR were substantially increased (Appendix A).

The Ki67 expression in T cells was detected using immunofluorescence assays, indicating that compared with the CD8^+^T cell group, there was no significant difference in the Ki67 expression among the M1 + CD8^+^T cell group and OE-TCF21 + TAM + CD8^+^T cell group and that in the M2 + CD8^+^T cell group and TAM + CD8^+^T cell group was greatly reduced, proving that the overexpression of TCF21 could promote the increase of T cell viability (Figure 7A). The results of ELISA showed that, compared with the CD8^+^T cell group, there was no significant difference in the content of cytokine IFN-γ in the M1 + CD8^+^T cell group and OE-TCF21 + TAM + CD8^+^T cell group. Conversely, the content of cytokine IFN-γ decreased in both the M2 + CD8^+^T cell group and TAM + CD8^+^T cell group with a significant difference (Figure 7D). The above results confirmed that overexpression of TCF21 promoted the polarization of TAMs to M1 and facilitated the viability of T cells by macrophages.

Apoptosis detection of tumor cells using flow cytometry (Figure 7B). Compared with the CD8^+^T cell group, the absolute percentage of trace-labeled tumor cells revealed no significant difference in the M1 + CD8^+^T cell group and OE-TCF21 + TAM + CD8^+^T cell group, indicating that the M1 + CD8^+^T cell group and OE-TCF21 + TAM + CD8^+^T cell group produced no significant impact on tumor cell killing vs. the CD8^+^T cell group. Meanwhile, the absolute percentage of trace-labeled tumor cells was increased both in the M2 + CD8^+^T cell group and TAM + CD8^+^T cell group with a significant difference, and both groups were weakened in killing tumor cells compared with the CD8^+^T cell group a significant effect (Figure 7C).

### 3.5. TCF21 Regulates the Immunosuppressive Effect of TAM Macrophages by Acting on Notch4

The results of qPCR and WB detection revealed that the expression of TCF21, Notch4, Hes1, and Hey1 in the OE-TCF21 group was markedly higher than that of the NC group, whereas the expression of Notch4, Hes1, and Hey1 was greatly decreased in the OE-TCF21 + si-Notch4 group versus the OE-TCF21 group (Figure 8A,B).

Compared with the NC group, expressions of TNF-α, IL-1β, and p40 mRNAs in the OE-TCF21 group were substantially elevated, while those of MRC1, FN1, and COX2 were greatly decreased. Compared to the OE-TCF21 group, expressions of TNF-α, IL-1β, and p40 mRNAs in the OE-TCF21 + si-Notch4 group were greatly decreased, whereas those of MRC1, FN1, and COX2 mRNAs were substantially increased (Figure 8C).

The results of flow cytometry indicated an increase in the positive expression percentages of CD86 and HLADR in the OE-TCF21 group versus NC, whereas those of CD163 and CD206 were decreased, and a significant difference was revealed; meanwhile, those of the CD86 and HLADR in OE-TCF21 + si-Notch4 group were decreased compared with the OE-TCF21 group whereas those of CD163 and CD206 increased, and the difference was significant. These findings demonstrated that over-expression of TCF21 promoted TAM polarization to M1 by acting on Notch4 (Appendix A).

### 3.6. The Exploration of the Direct Interaction between TCF21 and Notch4

Dual-luciferase reporter assays demonstrated the presence of direct interaction between TCF21 and Notch4 (Figure 9A). ChIP experiments showed that TCF21 was detected in both the IP group and Input group, indicating that the pull-down product was a TCF21 antibody-specific pull-down product, which was applicable for subsequent qPCR detection (Figure 9B). qPCR assays were performed to detect the binding site sequence in the pull-down product, and the results confirmed that Notch4 was the downstream target gene of the transcription factor TCF21 because P1 content was detected the highest in the pull-down product, indicating that P1 was the main binding site of TCF21 on the Notch4 promoter (Figure 9C).

## 4. Discussion

In recent years, with the change in living environment and lifestyle, the incidence of global lung cancer has increased year by year, and it has now become the most frequently diagnosed malignant tumor in the world. NSCLC is the most common type of lung cancer, accounting for about 85% of all lung cancer cases. A majority of NSCLC patients in China are diagnosed at an advanced stage due to the high invasiveness of the disease’s pathological characteristics as well as the lack of definite and effective early screening schemes. They have unfortunately missed the optimal opportunity to accept the surgical management of this condition. It is of vital importance to conduct in-depth investigations on the occurrence, development, and new strategies of diagnosis and treatment against NSCLC. This study analyzed NSCLC-related DEGs via the TCGA database using a bioinformatics approach, and significantly down-regulated genes TCF21 and Notch4 were screened out.

As a tumor suppressor, TCF21 exerts an essential role in the development of tumors, and it is considered to be a key regulator of invasion and metastasis of multiple malignant tumors [11]. Among NSCLC patients, the expression of TCF21 was substantially regulated by the methylation level of TCF21 and was intimately related to tumor staging and tumor metastasis [12]. The Notch signaling pathway is an important form of intercellular communication, which regulates cell development and differentiation, plays a role in cell repair and homeostasis maintenance, and participates in the regulation of tumor angiogenesis [13]. In mammals, the Notch pathway has been identified to consist of four Notch receptors (Notch 1–4) and five ligands [14]. Some studies have reported that the receptors and ligands of the Notch pathway are dysregulated in NSCLC and other human malignancies [15,16,17]. The Notch signaling pathway not only promotes the proliferation of NSCLC cells but also mediates the metastasis of NSCLC [18]. Notch4 is expressed in tumor cells, and it affects tumor invasion, metastasis, and patient prognosis [19]. Numerous studies have demonstrated that the Notch signaling pathway plays a crucial role in macrophage polarization. When the Notch signaling is inhibited, macrophages primarily exhibit the M2 phenotype. Activation of the Notch signaling can induce macrophage polarization from the M2 phenotype to the M1 phenotype, enhancing their anti-tumor capabilities [20]. This study demonstrated that TCF21 and Notch4 were greatly down-regulated in lung cancer cells and TAMs, and Notch4 was the downstream target gene of TCF21.

Meanwhile, macrophages, as a type of innate immune cells, participated in the body’s antiviral and anti-tumor effects. Under the context of the tumor microenvironment, the macrophages derived from peripheral blood are able to enter tumor tissues, making it possible to form TAMs infiltrating tumor tissue. TAMs are an important component of tumor inflammatory microenvironment, which can promote tumor invasion, metastasis, and angiogenesis [21]. TAMs can be polarized by cytokines and chemokines secreted by tumor cells and tumor stromal cells, allowing them to form two phenotypes with opposite characteristics and functions. One is the M1 type with pro-inflammatory and anti-tumor activities [22]; the other is the M2 type macrophages formed due to polarization, which exert roles in anti-inflammation, tumorigenesis, growth, and metastasis [23]. Commonly used marker molecules for M1 macrophages include IL-6, IL-1β, CD86, and HLADR, while those for M2 macrophages include IL-10, CD163, CD206, and others [24]. Advanced lung cancer cells are infiltrated with a great deal of M2-polarized macrophages, and the TAM-targeting antibody can substantially reduce lung tumor metastasis after acting on lung cancer mice [25,26]. The current research confirmed that the expression of TCF21, Notch4, Hes1, Hey1, TNF-α, IL-1β, and p40 in M2 macrophages and TAMs was much lower than that in M1 macrophages, whereas the expression of MRC1, FN1, and COX2 was remarkably higher than that of M1 macrophages. Study has shown that TCF21 is positively correlated with M2 macrophages [27]. This study also demonstrated that TCF21 promotes the polarization of macrophages towards M2.

Typical target genes of Notch signaling transduction include Hes and Hey [28]. It has been proven that Notch signaling transduction is related to anti-tumor immunotherapy [29]. TNF-α is a pro-inflammatory cytokine that can be secreted by TAM and cancer cells. Meanwhile, TNF-α is also intimately correlated to the prognosis of NSCLC patients in the promotion of inflammation, epithelial–mesenchymal transition, invasion, and metastasis under a tumor microenvironment. IL-1β is mainly synthesized and secreted by monocyte-macrophages and plays an important role in body immune regulation [30]. Being a frequently applied tumor marker, P40 is usually expressed in the basal layer of epithelial tissues, and its expression is up-regulated in squamous cell carcinoma tissues [31]. MRC1 is an M2-specific marker gene and is highly expressed in M2 macrophages [32]. FN1 is expressed in stromal cells but is mainly expressed in TAMs when in immune cells [33]. The overexpression of COX-2 is related to tumorigenesis and inflammatory response [34]. The results of the present experiment revealed that M1- and M2-type polarization were successfully induced.

In recent years, the discovery of immune checkpoint blockers targeting lung tumors has effectively demonstrated the immunogenicity of NSCLC [35]. Although immune checkpoint blockers are widely used in the treatment of advanced NSCLC, most patients develop innate or acquired immune resistance during immunotherapy. Therefore, exploring new molecular targets that may reverse or stop immune escape will help further improve the prognosis and survival of patients with NSCLC. The immune memory generated during immunotherapy could achieve long-term protection for the body, effectively preventing tumor recurrence and metastasis, which is expected to completely cure the tumor. The polarization of macrophages exerts an important role in tumor immunotherapy. TAMs can support tumor development by interacting with T cells [36]. First, macrophages polarize to the M1 phenotype, making it possible to transform from a state of promoting tumor suppressive immunity to a state of anti-tumor immunity promotion, thereby creating a pro-inflammatory tumor microenvironment and improving the inhibitory effect of TME on immunotherapy [37]. M2-type TAMs produce immunosuppressive factors, thereby inhibiting the function of CD4^+^ and CD8^+^T cells. Meanwhile, it secretes chemokines to inhibit the anti-tumor response of the tumor immune microenvironment. The combination of the regulation of macrophage phenotype and tumor immunotherapy can perfectly improve the efficiency of immunotherapy [38]. For instance, Kim et al. have enhanced the anti-tumor effect of the immune checkpoint inhibitor PD-L1 by polarizing M2-type macrophages in TAMs to M1-type [39], while Huang et al. applied the polarization of TAMs to enhance the treatment benefits of tumor vaccines [40]. This experiment demonstrated that the polarization of TAM to M1 can promote the viability of CD8^+^T cells, while the polarization of TAM to M2 reduces the viability of CD8^+^T cells and tumor killing. Overexpression of TCF21 promotes the polarization of TAM to M1 and the increase of T cell viability by acting on the Notch signaling pathway. Certain studies have indicated a close relationship between Notch signaling transduction and macrophage polarization [41]. Overall, our findings demonstrated that the downregulation of TCF21 in NSCLC might result in an enhanced immunosuppressive effect of TAM, which is not conducive to tumor management.

## 5. Conclusions

Our current research offered a detailed examination of the tumor microenvironment, providing valuable insights for guiding clinical immunotherapy effectively. Additionally, our investigation into the regulation of tumor-associated macrophage polarization opened up new avenues for the development of novel drugs. In this study, we validated that TAM polarization significantly impacts the viability of immune cells. Specifically, M1 polarization has the potential to reinstate macrophages’ anti-tumor functionality, alleviate the immunosuppressive nature of the tumor microenvironment, and enhance the effectiveness of immunotherapy. Meanwhile, overexpression of TCF21 affected the immunosuppressive effect of macrophages through the Notch signaling pathway.

## Figures and Tables

**Figure 1 pharmaceutics-15-02295-f001:**
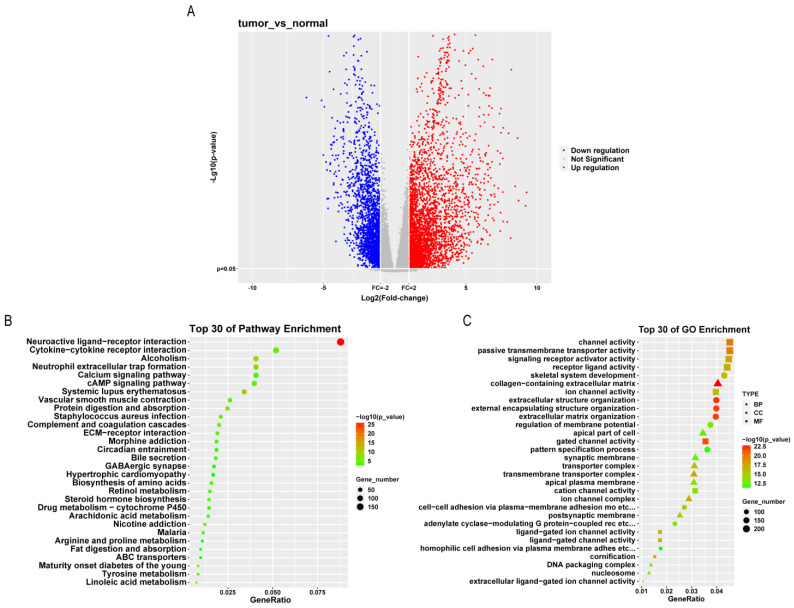
Screening and enrichment analysis of DEG. (**A**) DEG volcano map. (**B**) GO enrichment analysis. (**C**) KEGG enrichment analysis.

**Figure 2 pharmaceutics-15-02295-f002:**
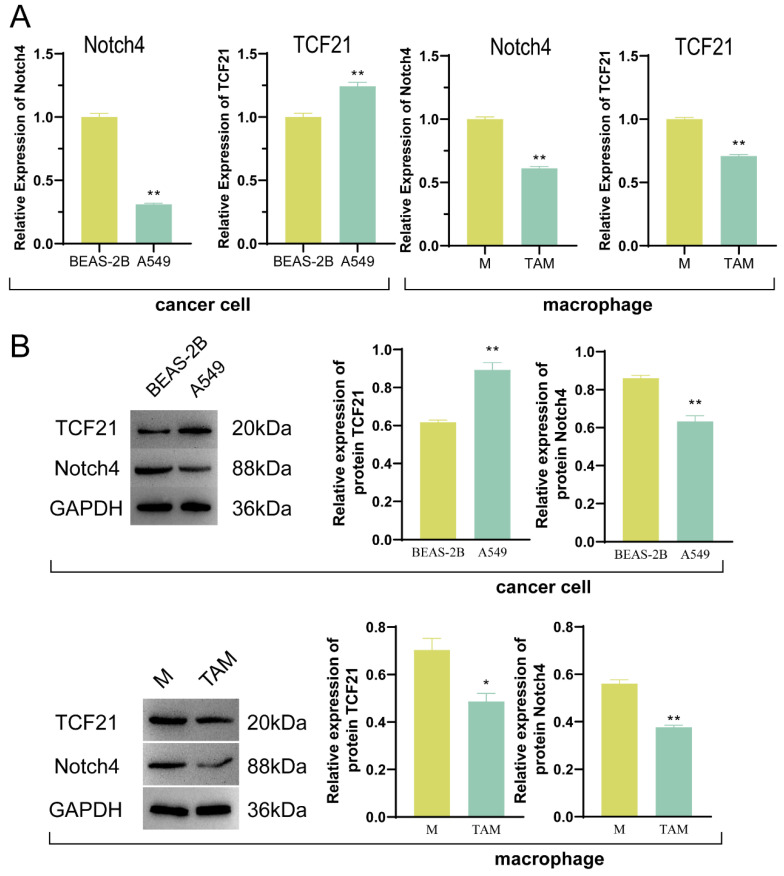
The expression of TCF21 was up-regulated, and Notch4 was down-regulated in A549 cells, while TCF21 and Notch4 expression were down-regulated in TAM compared with M. (**A**) qPCR detection of TCF21 and Notch4 mRNA expression. (**B**) The expression of TCF21 and Notch4 proteins was detected by Western blot. Note: * *p* < 0.05 and ** *p* < 0.01 represents a significant statistical difference.

**Figure 3 pharmaceutics-15-02295-f003:**
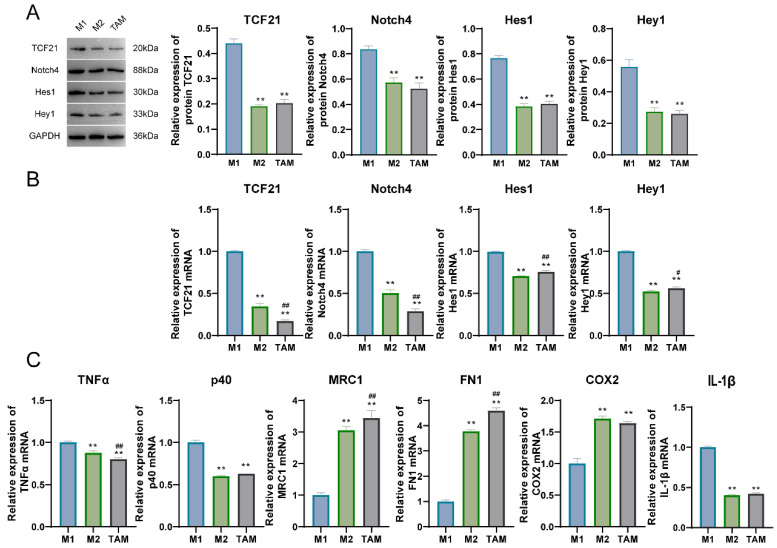
The expression levels of TCF21, Notch4, Hes1, Hey1, TNF-α, p40, and IL-1β were decreased in M2 and TAM, while the expressions of MRC1, FN1, and COX2 were increased compared with M1. (**A**) The expression level of TCF21, Notch4, Hes1, and Hey1 proteins was detected by Western blot. (**B**) The expression level of TCF21, Notch4, Hes1, and Hey1 mRNAs was detected by qPCR. (**C**) The expression level of TNFα, P40, MRC1, FN1, COX2, and IL-1β mRNAs was detected by qPCR. Note: ** *p* < 0.01 represents a significant statistical difference vs. M1; # *p* < 0.05 ## *p* < 0.01 represent a significant statistical difference vs. M2.

**Figure 4 pharmaceutics-15-02295-f004:**
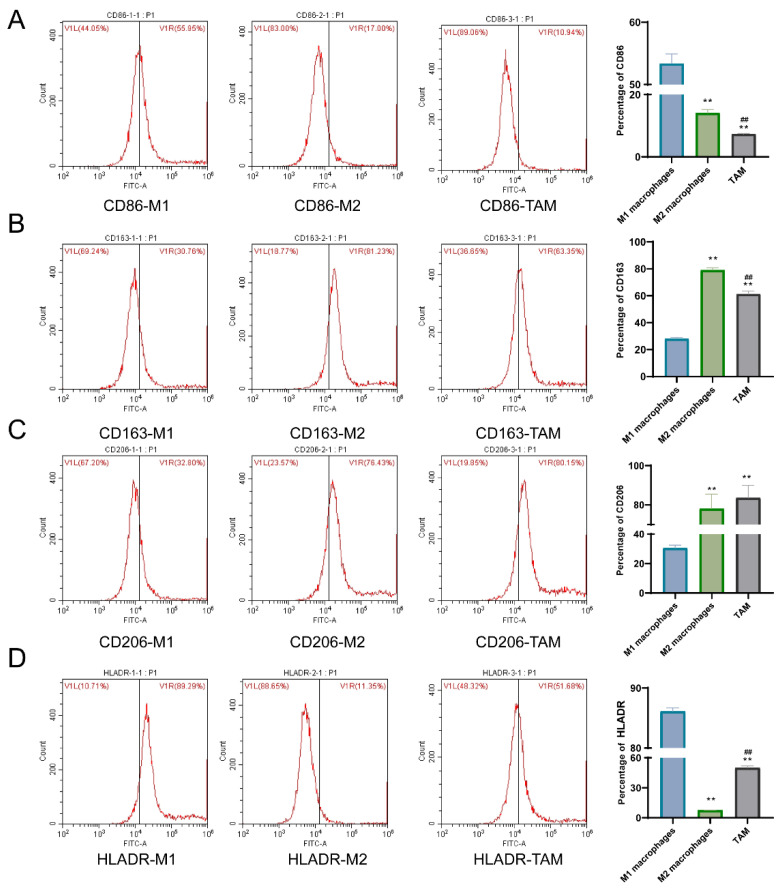
The positive expression rates of CD86 and HLADR in M2 macrophages and TAM groups were decreased, while the positive expression rates of CD163 and CD206 were increased compared with the M1 macrophages group. (**A**) Percentage of CD86 positive expression. (**B**) Percentage of CD163 positive expression. (**C**) Percentage of CD206 positive expression. (**D**) Percentage of HLADR positive expression. Note: ** *p* < 0.01 represents a significant statistical difference vs. M1 macrophages; ## *p* < 0.01 represents a significant statistical difference vs. M2 macrophages.

**Figure 5 pharmaceutics-15-02295-f005:**
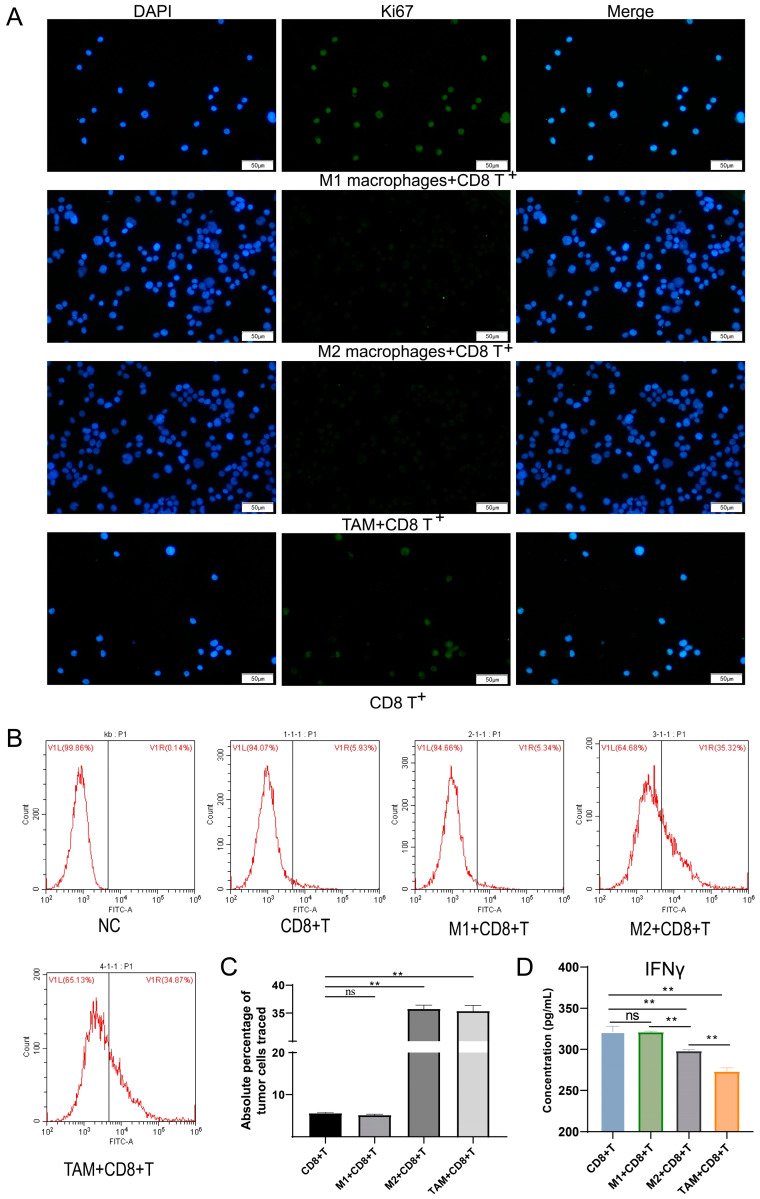
Impacts of macrophage polarization on T cell viability and tumor killing. (**A**) Expression of cytokine Ki67 in the M2 + CD8^+^T cell group and TAM + CD8^+^T cell group was greatly decreased, implying that TAMs polarized to the M2 type and T cell viability was weakened. (**B**,**C**) Flow cytometry result showed that compared with the CD8^+^T cell group, the M2 + CD8^+^T cell group and TAM + CD8^+^T cell group was weakened in killing tumor cells and produced a significant effect. (**D**) ELISA result showed the content of cytokine IFN-γ in the M2 + CD8^+^T cell group and TAM + CD8^+^T cell group decreased. Note: ns represente no significant difference; ** *p* < 0.01 represents a significant statistical difference.

**Figure 6 pharmaceutics-15-02295-f006:**
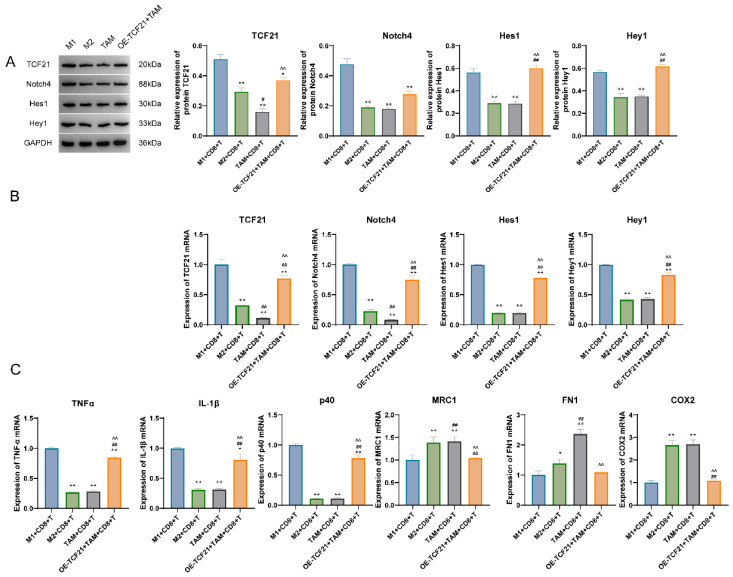
Impacts of TCF21 in macrophages on T cell viability and tumor killing. (**A**) The expression of TCF21, Notch4, Hes1, and Hey1 proteins detected by Western blot. (**B**) The expression of TCF21, Notch4, Hes1, and Hey1 mRNAs was detected by qPCR. (**C**) The expression of TNFα, P40, MRC1, FN1, COX2, and IL-1β mRNAs was detected by qPCR. Note: * *p* < 0.05 and ** *p* < 0.01 represent a significant statistical difference vs. M1 + CD8^+^T; ## *p* < 0.01 represents a significant statistical difference vs. M2 + CD8^+^T; ^^ *p* < 0.01 represents a significant statistical difference vs. TAM + CD8^+^T.

**Figure 7 pharmaceutics-15-02295-f007:**
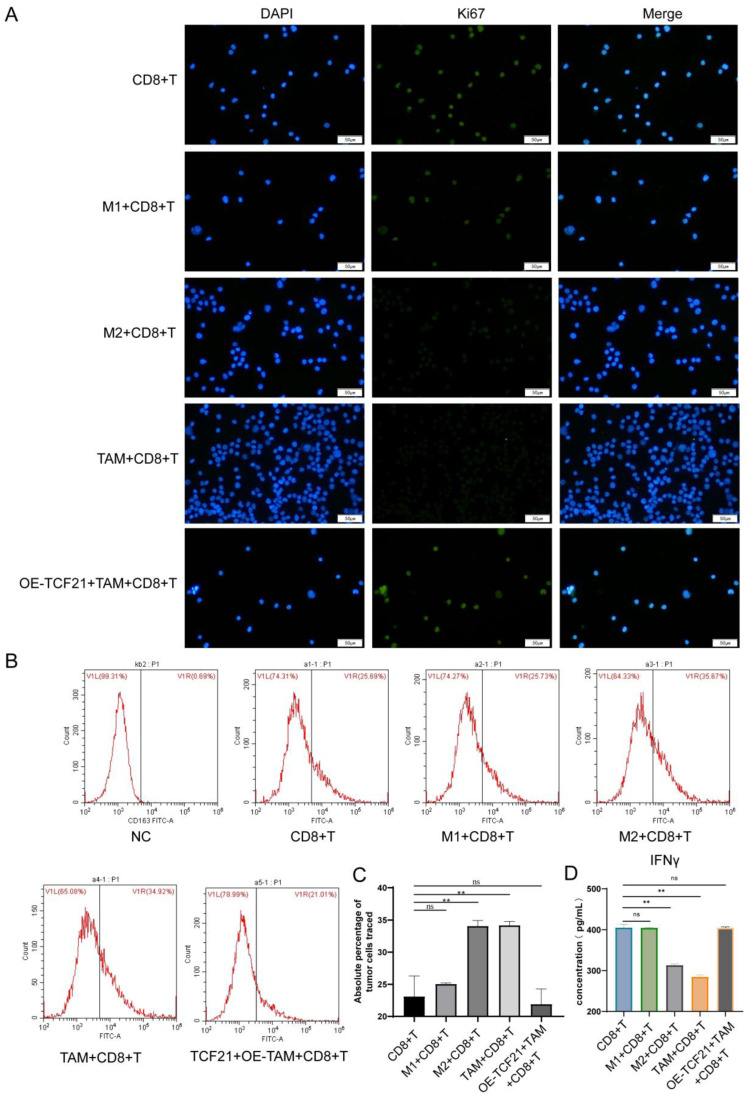
TCF21 mediated the influence of macrophages on T cell viability and tumor killing. (**A**) Immunofluorescence results showed that the overexpression of TCF21 could promote the increase of T cell viability in TAM. (**B**,**C**) Flow cytometry results indicated that overexpression of TCF21 reduced the percentage of tumor cells in TAM. (**D**) ELISA results proved that overexpression of TCF21 promoted the increase of INF-γ content in TAM. Note: ns representes no significant difference; ** *p* < 0.01 represents a significant statistical difference.

**Figure 8 pharmaceutics-15-02295-f008:**
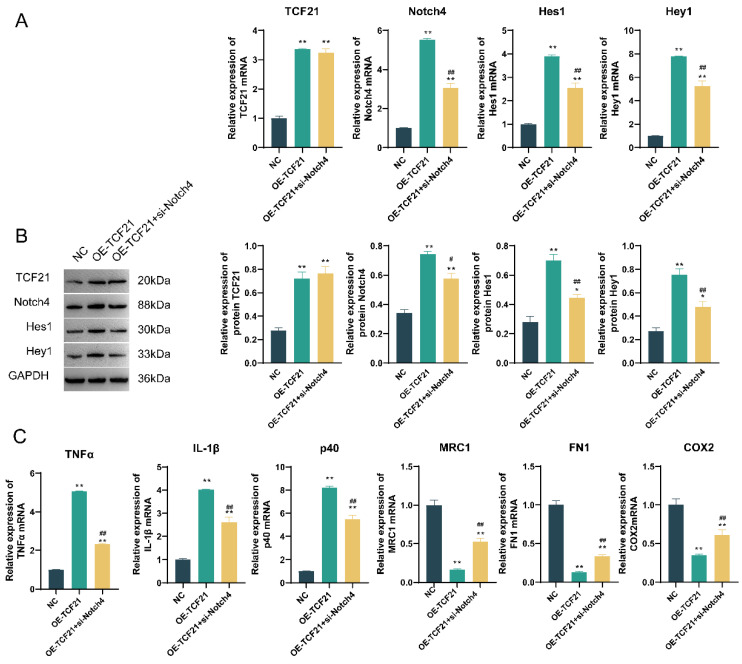
The regulatory effect of TCF21 on Notch4. (**A**) The expression of TCF21, Notch4, Hes1, and Hey1 mRNAs was detected by qPCR. (**B**) The expression of TCF21, Notch4, Hes1, and Hey1 proteins was detected by Western blot. (**C**) TCF21 acts with Notch4 to regulate TNF-α, P40, MRC1, FN1, COX2, and IL-1β. Note: * *p* < 0.05 and ** *p* < 0.01 represent a significant statistical difference vs. NC; # *p* < 0.05 and ## *p* < 0.01 represent a significant statistical difference vs. OE-TCF21.

**Figure 9 pharmaceutics-15-02295-f009:**
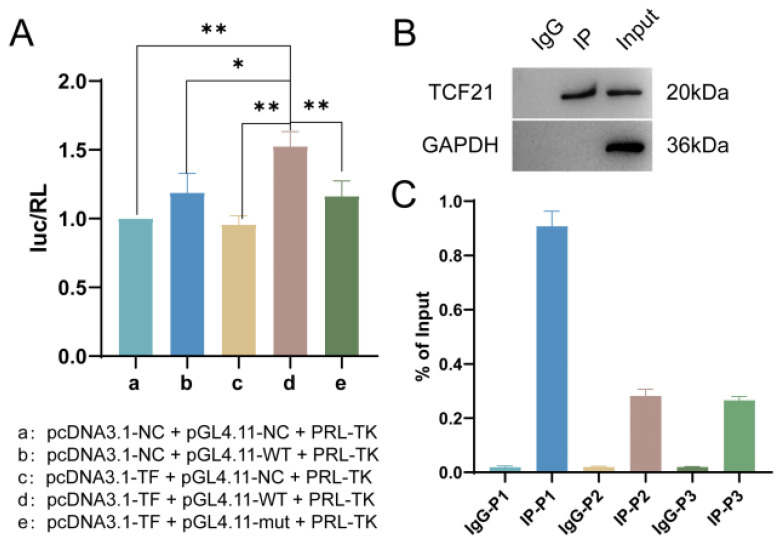
The presence of direct interaction between TCF21 and Notch4. (**A**) Dual-luciferase reporter assay. (**B**) CHIP assay. (**C**) CHIP-qPCR assay. Note: * *p* < 0.05 represents a statistical difference, ** *p* < 0.01 represents a significant statistical difference.

**Table 1 pharmaceutics-15-02295-t001:** Primers sequences.

Primers	Sequence 5′-3′
TCF21-F	CGTGACTGTCCCTCTGTGTC
TCF21-R	ATGCTGGCATTGCTCGTGG
Notch4-F	CGCTACTATGAGCGACGGTT
Notch4-R	GCGTGCTGCTCCATGTTATG
Hes1-F	AGGACCGGGAGTAAATTGCAG
Hes1-R	CCTTTCCCAGACTCGCACCT
Hey1-F	TACAGTCCGGGACCTTCCAA
Hey1-R	AGGGACCTCAGTGTGTGCTA
JNK-F	TACAGTCCGGGACCTTCCAA
JNK-R	AGGGACCTCAGTGTGTGCTA
TNFα-F	ACACCATGAGCACTGAAAGC
TNFα-R	TCCATACACACTTAGTGAGCACC
IL-1β-F	TCCCAGTCCTATCCCTCGTG
IL-1β-R	CGCGCTTGGAATGCTTGTTA
p40-F	GGACAACTGCACCAGACCAT
p40-R	AGGGTTGGCATCACCATTCAC
MRC1-F	GCATCCCACAGCATTAGCAC
MRC1-R	CGGCTCTCATGGTGCATCTA
FN1-F	TTAGCGCCTGGCTGTTGTAT
FN1-R	TGTGCATACAGACCAGTTCCA
COX2-F	CGTCTGAACTATCCTGCCCG
COX2-R	GTCGTGTAGCGGTGAAAGTG

## Data Availability

The data used to support the findings of this study are included within the article.

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
