# Peer review of "Mechanism of TCF21 Downregulation Leading to Immunosuppression of Tumor-Associated Macrophages in Non-Small Cell Lung Cancer"

_pharmaceutics, 2023, doi:10.3390/pharmaceutics15092295_

Round 1
Reviewer 1 Report
The presented manuscript covers a thorough investigation of the effect of TCF21 on macrophage polarizations and corresponding consequences when cocultured with CD8+ T cells. In this manuscript, the authors conducted in vitro studies to prove the potential of TCF21 regulation to treat immunosuppressive lung cancers. Overall, this is a comprehensive research article with sufficient data that would attract interest in this field. Here attached my comments.
Major points:
1. There is abundant literature studying the markers (CD86, CD163, CD206 and HLA-DR) on macrophage subtypes. These are markers to distinguish macrophage subtypes. The effect of macrophage polarization on CD8+ T cells is also well-reported. I suggest the authors focus on the effect of TCF21 pathways in macrophage polarization, explaining more details regarding the importance/rationale of the genes/cytokines investigated in this manuscript. The macrophage markers distinguishment could be used as the proof of successful development of macrophage subtypes.
2. Some sections of the methods need rewriting. For example, in section 2.2, “The macrophage density of A549 cells…”. Also, the protocol to develop M1, M2, and TAM needs to be validated either by the authors or in the literature (please cite the corresponding reference). In section 2.8, missing words in the first sentence. Please provide the source of CD8+ T cells.
Minor points
3. Double-check author information.
4. All the fluorescence imaging figures need to be higher resolution, please replace these figures (Fig 6 and Fig 10).
5. There are too many figures in this manuscript. The raw data of flow cytometry could be moved to supplemental files and combined with some of the flow cytometry results.
6. I’m wondering whether the authors consider using primary macrophages for this study.
7. Explain what the OE-TCF21 is in the results section.
Moderate editing of English language required. There are some missing words and incomplete sentences.
Author Response
Major points:
- There is abundant literature studying the markers (CD86, CD163, CD206 and HLA-DR) on macrophage subtypes. These are markers to distinguish macrophage subtypes. The effect of macrophage polarization on CD8+ T cells is also well-reported. I suggest the authors focus on the effect of TCF21 pathways in macrophage polarization, explaining more details regarding the importance/rationale of the genes/cytokines investigated in this manuscript. The macrophage markers distinguishment could be used as the proof of successful development of macrophage subtypes.
R: Thank you for your valuable advice, which will help us to further improve this study. We have added the introduction of markers related to macrophage polarization, and added the introduction of TCF21 and its correlation.
“Meanwhile, macrophages, as a type of innate immune cells, participated in the body's antiviral and antitumor effects. Under the context of tumor microenvironment, the macrophages derived from peripheral blood are able to enter tumor tissues making it possible to form TAMs infiltrating tumor tissue. TAMs are an important component of tumor inflammatory microenvironment, which can promote tumor invasion, metastasis and angiogenesis[21]. TAMs can be polarized by cytokines and chemokines secreted by tumor cells and tumor stromal cells, allowing to form two phenotypes with opposite characteristics and functions. One is the M1 type with pro-inflammatory and anti-tumor activities[22]; The other is the M2 type macrophages formed due to polarization which exert roles in anti-inflammation, tumorigenesis, growth, and metastasis[23]. The marker molecules commonly used in M1 macrophages are IL-6, IL-1β, CD86 and HLADR, and the marker molecules commonly used in M2 macrophages are IL-10, CD163, CD206 and so on[24]. Advanced lung cancer cells are infiltrated with a great deal of M2-polarized macrophages, and the TAM-targeting antibody can substantially reduce lung tumor metastasis after acting on lung cancer mice[25,26]. The current research confirmed that the expression of TCF21, Notch4, Hes1, Hey1, TNF-α, IL-1β and p40 in M2 macrophages and TAMs was much lower than that in M1 macrophages, whereas the expression of MRC1, FN1 and COX2 was remarkably higher than that of M1 macrophages. Study has shown that TCF21 is positively correlated with M2 macrophages[27]. This study also demonstrated that TCF21 pro-motes the polarization of macrophages towards M2.”
- Some sections of the methods need rewriting. For example, in section 2.2, “The macrophage density of A549 cells…”. Also, the protocol to develop M1, M2, and TAM needs to be validated either by the authors or in the literature (please cite the corresponding reference). In section 2.8, missing words in the first sentence. Please provide the source of CD8+ T cells.
Reply: Thanks for your advice, the density of A549 cells, the corresponding reference and the source of CD8+ T cells have been supplemented, as follows:
“TAM group: The A549 cells in good growth state were digested and harvested using the same method as passage, adjust the density of A549 cells and PMA induced macrophages for 5 days to 1x10^ 5 cells/mL, and the cells were inoculated in the upper chamber of the Transwell, and the macrophages induced by PMA were inoculated in the lower chamber of the Transwell and co-cultured in equal proportions for 48 h.”
“The literature related to the polarization of M1, M2, and TAM has been inserted after subsection 2.3. as follows:
[8] Zhang Y, Chen Y, Ding T, et al. Janus porous polylactic acid membranes with versatile metal-phenolic interface for biomimetic periodontal bone regeneration. NPJ Regenerative Medicine 2023, 8, 28.
[9] Yuan S, Dong Y, Peng L, et al. Tumor-associated macrophages affect the biological behavior of lung adenocarcinoma A549 cells through PI3K/AKT signaling pathway. Oncology Letters 2019, Issue 18, 1840-1846. "
"2.8. Immunofluorescence assay
First fix the cell slides, add an appropriate amount of 0.3% Triton X-100 permeabilization solution, and incubate at room temperature for 5 min.
"2.4. Effect of Experimental Cell Grouping on T Cell Viability and Macrophage Polarization for Tumor Killing
CD8+ T cells were isolated and purified using MACSxpress Whole Blood CD8 T Cell Isolation Kit (130-098-194, Novo Biotechnology Co., Ltd.), and identified by flow cytometry.
Minor points
- Double-check author information.
Reply: Thank you for your reminder, the author information is correct.
- All the fluorescence imaging figures need to be higher resolution, please replace these figures (Fig 6 and Fig 10).
Reply: Thank you for your comments on our articles. It is possible that our pictures are compressed in the article, resulting in a loss of clarity, so we will provide a separate file.
- There are too many figures in this manuscript. The raw data of flow cytometry could be moved to supplemental files and combined with some of the flow cytometry results.
Response: Thanks for your suggestion, the figures have been merged into 9 and some of the flow cytometry results have been moved to Supplementary Files.
- I’m wondering whether the authors consider using primary macrophages for this study.
R: We are not using primary macrophages, but cells that go through them.
- Explain what the OE-TCF21 is in the results section.
Reply: OE represents overexpression, the meaning of OE-TCF21 has been explain in the result section.
“3.4. Impacts of TCF21 in macrophages on T cell viability and tumor killing
The results of qPCR and WB detection showed that the expression of TCF21, Notch4, Hes1 and Hey1 were markedly decreased in the M2 + CD8+T cell group and TAM + CD8+T cell group as compared to M1 + CD8+T cell group; The expression of the described cytokines in the TAM + CD8+T cell group was greatly lower than that in the overexpression (OE)-TCF21 + TAM + CD8+T cell group (Figure 8A-B).”
Comments on the Quality of English Language
Moderate editing of English language required. There are some missing words and incomplete sentences.
Reply: Thank you very much for your careful review of this article, which will help us to further improve this research. We have improved the language of this article based on your suggestions.

Reviewer 2 Report
Dear Authors,
Re: pharmaceutics-2507499 Title: "Mechanism of TCF21 downregulation leading to immunosuppression of tumor-associated macrophages in non-small cell lung cancer"
Your manuscript describes a study aimed to explore mechanisms of TCF21 down regulation causing the immunosuppressive effect of TAMs in non-small cell lung cancer (NSCLC), in order to provide new targets for the immunotherapy of NSCLC.
Please find my comments below:
1. In the Abstract the abbreviations TCF21, M1 and TAM are not described.
2. Please correct the punctuation in the abstract: "... cells; Meanwhile, ..." and check for similar errors in the manuscript.
3. Please provide more information and discussion about the relation between TCF21 and the Notch signalling pathway, specifically Notch4.
4. Why did you apply 50 ng/mL of PMA for a 5 days period? A related reference is needed.
5. Please add the GSE number or Dataset ID related to NSCLC data from the TCGA database to the Methods section.
6. In the Methods section, the enrichment method has not been mentioned. Please add it.
7. It would have been better if you had used more advanced methods such as WGCNA or single-cell analysis for the bioinformatic results' validation before discussing.
8. It's better to provide a bit more explanation, especially regarding the signalling pathways involved in this mechanism.
Author Response
- In the Abstract the abbreviations TCF21, M1 and TAM are not described.
Reply: Thanks for your suggestion, the full name of the abbreviations TCF21, M1 and TAM have been added to the abstract.
“Abstract: Lung cancer, as one of the high-mortality cancers, seriously affects the normal life of people. Non-small cell lung cancer (NSCLC) accounts for a high proportion of the overall incidence of lung cancer, and identifying therapeutic targets of NSCLC is of vital significance. This study attempted to elucidate the regulatory mechanism of transcription factor 21 (TCF21) on the immunosuppressive effect of tumor-associated macrophages in NSCLC. The experimental results revealed that the expression of TCF21 was decreased in lung cancer cells and tumor-associated macrophages; macro-phage polarization affected T cell viability and tumor-killing greatly, and M2 type polarization reduced the viability and tumor killing of CD8+T cells; Meanwhile, overexpression of TCF21 promoted the polarization of tumor-associated macrophage (TAM) to M1 macrophages, and the enhancement of macrophages to the viability of T cells; Further, TCF21 might have a targeting relationship with Notch, and TCF21 might play its role through the Notch signaling pathway. This study demonstrated the polarization regulation of tumor-associated macrophages to regulate the immunosuppressive effect, which provides novel targets for the treatment of lung cancer.”
- Please correct the punctuation in the abstract: "... cells; Meanwhile, ..." and check for similar errors in the manuscript.
Reply: Thanks for your comments. The punctuation in the abstract: "... cells; Meanwhile, ..." has been revised, and the article has been checked.
“Abstract: Lung cancer, as one of the high-mortality cancers, seriously affects the normal life of people. Non-small cell lung cancer (NSCLC) accounts for a high proportion of the overall incidence of lung cancer, and identifying therapeutic targets of NSCLC is of vital significance. This study attempted to elucidate the regulatory mechanism of transcription factor 21 (TCF21) on the immunosuppressive effect of tumor-associated macrophages in NSCLC. The experimental results revealed that the expression of TCF21 was decreased in lung cancer cells and tumor-associated macrophages. Macrophage polarization affected T cell viability and tumor-killing greatly, and M2-type polarization reduced the viability and tumor-killing of CD8+T cells. Meanwhile, overexpression of TCF21 promoted the polarization of tumor-associated macrophages (TAM) to M1 macrophages, and the enhancement of macrophages to the viability of T cells. Further, TCF21 might have a targeting relationship with Notch, and TCF21 might play its role through the Notch signaling pathway. This study demonstrated the polarization regulation of tumor-associated macrophages to regulate the immunosuppressive effect, which provides novel targets for the treatment of lung cancer.”
- Please provide more information and discussion about the relation between TCF21 and the Notch signaling pathway, specifically Notch4.
R: Thank you very much for your constructive comments on this study, and we agree with your suggestions very much. However, we searched the extensive literature and found few studies showing a correlation between TCF21 and the Notch signaling pathway. Therefore, in this study, we only discussed their respective roles in NSCLC.
- Why did you apply 50 ng/mL of PMA for a 5 days period? A related reference is needed.
R: The application of 50 ng/mL of PMA (phorbol 12-myristate 13-acetate) for a 5-day period is a widely used protocol to induce the differentiation of THP-1 cells into macrophage-like cells. PMA acts as an activator of protein kinase C (PKC), leading to downstream signaling events that drive the differentiation process. The high concentration (50 ng/mL) of PMA is necessary to robustly activate PKC and initiate the signaling cascades that trigger cellular differentiation. A 5-day duration of PMA treatment is often chosen based on previous studies and optimization experiments. During this period, the THP-1 cells undergo morphological and functional changes similar to those observed in differentiated macrophages. These changes include adherence to the culture dish, altered cell shape, enhanced phagocytic ability, and increased expression of macrophage-specific markers. The related reference has been added.
“Group M: The cells were induced and cultured in a RPMI-1640 medium containing 50 ng/mL PMA for 5 days[8].”
[8] Zhang X, Wang G, Gurley EC, Zhou H. Flavonoid apigenin inhibits lipopolysaccharide-induced inflammatory response through multiple mechanisms in macrophages. PloS One 2014, 9, e107072.
- Please add the GSE number or Dataset ID related to NSCLC data from the TCGA database to the Methods section.
Reply: The data set related to NSCLC in the TCGA database used in our article is "TCGA-LUAD". This content has been added to the method.
“2. Materials and methods
2.1. Differentially expressed gene (DEG) analysis
Relevant NSCLC data were downloaded from the TCGA database, the dataset TCGA-LUAD. The R package DEseq2 was applied to analyze DEGs of the samples. After the p-value was calculated, multiple hypothesis testing was used for correction. The threshold of the p-value was determined by controlling the false discovery rate (FDR) and the corrected p-value was used as the q-value. The differential expression fold was subsequently calculated based on the FPKM value and expressed as Fold-change. The screening indicators for this analysis were p-value < 0.05, log2FC > 1 or < -1.”
- In the Methods section, the enrichment method has not been mentioned. Please add it.
Reply: the enrichment method has been added to Method section.
“2.1. Differentially expressed gene (DEG) analysis
Relevant NSCLC data were downloaded from the TCGA database, the dataset TCGA-LUAD. The R package DEseq2 was applied to analyze the DEGs of the samples. After the p-value was calculated, multiple hypothesis testing was used for correction. The threshold of the p-value was determined by controlling the false discovery rate (FDR) and the corrected p-value was used as the q-value. The differential expression fold was subsequently calculated based on the FPKM value and expressed as Fold-change. The screening indicators for this analysis were p-value < 0.05, log2FC > 1 or < -1. Enrichment analysis of the Gene Ontology (GO) and Kyoto Encyclopedia of Genes and Genomes (KEGG) path-ways for DEGs was performed using the clusterProfiler (v4.2.2) package.”
- It would have been better if you had used more advanced methods such as WGCNA or single-cell analysis for the bioinformatic results' validation before discussing.
R: We strongly agree with your suggestion, and the validation of more advanced methods will enhance the credibility of the results of this study. However, we believe that the cell experiment is highly reliable for the verification of bioinformatic results, so we have not conducted other verification. In future experiments, we will follow your suggestion to add more verification methods.
- It's better to provide a bit more explanation, especially regarding the signalling pathways involved in this mechanism.
R: Thank you very much for taking the time to carefully review our research and providing constructive feedback. Since no relevant studies have been found to prove the relationship between TCF21 and Notch signaling pathway, no discussion on the correlation between the two was provided in this study. However, we provide a discussion of the role of the Notch signaling pathway in non-small cell lung cancer and its relevance to tumor-associated macrophages.
“As a tumor suppressor, TCF21 exerts an essential role in the development of tumors, and it is considered to be a key regulator of invasion and metastasis of multiple malignant tumors[11]. Among NSCLC patients, the expression of TCF21 was substantially regulated by the methylation level of TCF21, and was intimately related to tumor staging and tumor metastasis[12]. The Notch signaling pathway is an important form of intercellular communication, which regulates cell development and differentiation, plays a role in cell repair and homeostasis maintenance, and participates in the regulation of tumor angiogenesis[13]. In mammals, this pathway has been identified four Notch receptors (Notch 1-4) and five ligands[14]. Some studies have reported that the receptors and ligands of the Notch pathway are dysregulated in NSCLC and other human malignancies[15-17]. The Notch signaling pathway not only promotes the proliferation of NSCLC cells, but also mediates the metastasis of NSCLC[18]. Notch4 is expressed in tumor cells and affects tumor invasion, metastasis and patient prognosis[19]. Numerous studies have demonstrated that the Notch signaling pathway plays a crucial role in macrophage polarization. When the Notch signaling is inhibited, macrophages primarily exhibit the M2 phenotype. Activation of the Notch signaling can induce macrophage polarization from the M2 phenotype to the M1 phenotype, enhancing their anti-tumor capabilities[20]. This study demonstrated that TCF21 and Notch4 were greatly downregulated in lung cancer cells and TAMs, and Notch4 was the downstream target gene of TCF21.”
Reviewer 3 Report
The authors have attempted to test the mechanism of TCF21 downregulation in NSCLC. While an interesting concept, the manuscript is lacking in many regards as follows-
1. The methods section is very poorly written with extremely vague descriptions and lack of appropriate terminologies. This makes it almost impossible to judge the validity of the results shown. For example, there are no names/ details about the different media used, cell seeding densities for various assays, quantification methods for western blots or fluorophores for any of the flow cytometry antibodies.
2. The text is very poorly written and organized with no clear rationale and hypothesis for any of the experiments performed. The introduction is generic and adds little to nothing towards setting up the research study for the readers.
3. Figures are missing legends for the most part and are not organized well. Much of the flow gating shown in figures 5,7 and 9 can be condensed into fewer plots with only one representative plot for the gating strategy
4. Discussion does not provide a clear overview of the context in which these data add to our knowledge base for NSCLC and role of tumor associated macrophages
Manuscript will benefit from English language editing services and requires a serious effort to add scientific details. In its current form, the language is far too lay for an audience of researchers and clinicians
Author Response
- The methods section is very poorly written with extremely vague descriptions and lack of appropriate terminologies. This makes it almost impossible to judge the validity of the results shown. For example, there are no names/ details about the different media used, cell seeding densities for various assays, quantification methods for western blots or fluorophores for any of the flow cytometry antibodies.
R: Thanks for your suggestion, the method section has been revised following your advice. The medium used in this study was RPMI-1640 medium and complete medium, and the corresponding names have been clearly indicated.
“2.2. Cell grouping treatment
Human lung cancer cells (A549) were purchased from BeNa Culture Collection. Healthy human lung epithelial cells (BEAS-2B) and human mononuclear cells (THP-1) were purchased from Procell Life Science&Technology Co., Ltd. The density of THP-1 cells was adjusted to 2×105 cells /mL and added to 6-well plates for culture at 37℃ in 5% CO2 cell incubator. THP-1 cells were randomly divided into two groups, M group and TAM group, and were treated as follows respectively. Group M: The cells were induced and cultured in a medium containing 50 ng/mL PMA for 5 days. TAM group: The A549 cells in good growth state were digested and harvested using the same method as passage, adjust the density of A549 cells and PMA induced macrophages for 5 days to 1x10^5 cells/mL, and the A549 cells were inoculated in the upper chamber of the Transwell, and the macrophages induced by PMA were inoculated in the lower chamber of the Transwell and co-cultured in equal proportions for 48 h.”
“2.6. Western blot assay
The tissue was lysed with RIPA lysate and the total protein was extracted, the bands were separated using electrophoresis, and the membrane was transferred at a constant current of 250 mA. Primary antibody was diluted with primary antibody diluent at 1:1000 and incubated overnight at 4℃. Secondary antibody was diluted to a certain concentration (1:2000) with blocking buffer and incubated for 1 h. The ECL exposure solution was mixed with liquid A and B at 1:1 rate, and then evenly covered on the entire membrane. After re-action for 1 min, it was loaded in the exposure meter for detection. The western blot bands were quantified using Image-J software (National Institutes of Health, Bethesda, MD, UAS). The primary antibodies used include TCF21 (abclonal, A17451); Notch4 (abclonal, A8303); Hes1 (abclonal, A0925); Hey1 (abclonal, A16110); GAPDH (abclonal, A19056); secondary antibody (abclonal, AS014).”
“2.7. Flow cytometry
The cells after centrifugation were gently blown, supplied with 100 μL of special fixative A (FIX & PERM), mixed well, and incubated at room temperature for 15 min in the dark. Of 1 mL PBS was added to the suspension after incubation, mixed well, and centrifuged at 1200 rpm for 5 min. The supernatant was discarded, and the operation was repeated once. The cells after centrifugation were gently blown, added with 100 μL of special membrane breaking agent B (FIX & PERM), mixed well, and incubated at room temperature in the dark for 20 min. After being added with 5 μL of corresponding antibodies to each tube, the cells were vibrated and mixed gently, the specific cytokines in the cells were stained, and incubated for 20 min at room temperature in the dark. Of 1 mL PBS was added to the suspension after incubation, mixed well, and centrifuged at 1200 rpm for 5 min. The supernatant was discarded, and the operation was repeated once. PBS at 100 μL was added to resuspend the cells and subsequently detected using flow cytometry. The percentage of apoptotic cells was determined by CytoFLEX flow cytometry (Beckman Coulter, Brea, CA, USA) and FlowJo V10 software. FIX&PERM Kit (MultiSciences (LiankeBio), China, GAS003/2, A10241); Anti-Human CD86 (Biolegend, USA, 374204, B270127); An-ti-Human CD206 (BD, USA, 551135,38855); Anti-Human CD163 (BD, USA, 563697, 57582); and Anti-Human HLADR (BD, USA, 555560, 37681).”
- The text is very poorly written and organized with no clear rationale and hypothesis for any of the experiments performed. The introduction is generic and adds little to nothing towards setting up the research study for the readers.
R: Thank you for your suggestion, which is more convincing for the rationality of the article research. We have added reasonable assumptions for this study based on your suggestions.
“Non-small cell lung cancer (NSCLC) is the most common histological subtype of lung cancer, accounting for 85% of the overall incidence of lung cancer, and it ranks first among male malignant tumors in China[1,2]. As the cancer cells of NSCLC spread slowly, and the early onset of patients lacks typical symptoms, a majority of patients are diagnosed at an advanced stage and have missed the best treatment period for the tumor, resulting in shortened period of survival[3]. Despite advancement has been achieved in clinical and experimental oncology in recent years, the 5-year survival rate of NSCLC is still at a low level[4]. Thus, exploring the molecular mechanism of NSCLC and identifying molecular markers related to NSCLC progression can produce remarkable clinical significance in improving the prognosis of patients.
Certain recent studies have indicated the importance of tumor-associated macrophages (TAMs) in the TME, NSCLC tumor progression, angiogenesis and distant metastasis[5], and they serve as an important factor in the prognosis of patients. It is hoped that through the in-depth study of TAMs, the microenvironment of the tumor matrix can be modified and reshaped to improve the therapeutic effect of drugs. TAMs can act in concert with tumor cells to promote tumor invasion and metastasis[6]. Taken together, the exploration of the regulation of TAM phenotype and its action mechanism on the development of lung adenocarcinoma is helpful for identifying new targets and novel drugs for the treatment of lung cancer.
Based on a bioinformatics analysis approach, this study screened out transcription factor 21 (TCF21), a markedly down-regulated gene in NSCLC. TCF21 is a lately discovered tumor suppressor gene, characterized by reversing epithelial-mesenchymal transition (EMT), and it is of great significance to the growth and differentiation of cells[7]. The role of TCF21 in NSCLC is unclear, and we hypothesize that TCF21 can lead to the immunosuppressive effects of TAM. It is therefore that this study aimed to explore the mechanism of TCF21 downregulation causing the immunosuppressive effect of TAMs in NSCLC, hoping to provide new targets for the immunotherapy of NSCLC.”
- Figures are missing legends for the most part and are not organized well. Much of the flow gating shown in figures 5,7 and 9 can be condensed into fewer plots with only one representative plot for the gating strategy.
R: Thanks for the reviewer’s helpful comments, the legends have been revised and the flow figures have removed excess flow gating, leaving only one representative plot for the gating strategy.
- Discussion does not provide a clear overview of the context in which these data add to our knowledge base for NSCLC and role of tumor associated macrophages
R: We sincerely appreciate your review comments on our work. The feedback you provide is crucial for us to further refine and improve our research. Therefore, I have supplemented and improved the discussion section of the article according to your suggestion.
“In recent years, with the change of living environment and lifestyle, the incidence of global lung cancer has increased year by year, and has now become the most frequently diagnosed malignant tumor in the world. NSCLC is the most common type of lung cancer, accounting for about 85% of all lung cancer cases. A majority of NSCLC patients in China are diagnosed at an advanced stage due to the high invasiveness of the disease patholog-ical characteristics as well as the lack of definite and effective early screening schemes, they have unfortunately missed the optimal opportunity to accept the surgical manage-ment of this condition. It is of vital importance to conduct in-depth investigations on the occurrence, development and new strategies of diagnosis and treatment against NSCLC. This study analyzed NSCLC-related DEGs via the TCGA database using a bioinformatics approach, and significantly downregulated genes TCF21 and Notch4 were screened out.
As a tumor suppressor, TCF21 exerts an essential role in the development of tumors, and it is considered to be a key regulator of invasion and metastasis of multiple malignant tumors[11]. Among NSCLC patients, the expression of TCF21 was substantially regulated by the methylation level of TCF21, and was intimately related to tumor staging and tumor metastasis[12]. The Notch signaling pathway is an important form of intercellular commu-nication, which regulates cell development and differentiation, plays a role in cell repair and homeostasis maintenance, and participates in the regulation of tumor angiogenesis[13]. In mammals, this pathway has been identified four Notch receptors (Notch 1-4) and five ligands[14]. Some studies have reported that the receptors and ligands of the Notch path-way are dysregulated in NSCLC and other human malignancies[15-17]. The Notch signaling pathway not only promotes the proliferation of NSCLC cells, but also mediates the metas-tasis of NSCLC[18]. Notch4 is expressed in tumor cells and affects tumor invasion, metasta-sis and patient prognosis[19]. Numerous studies have demonstrated that the Notch signal-ing pathway plays a crucial role in macrophage polarization. When the Notch signaling is inhibited, macrophages primarily exhibit the M2 phenotype. Activation of the Notch sig-naling can induce macrophage polarization from the M2 phenotype to the M1 phenotype, enhancing their anti-tumor capabilities[20]. This study demonstrated that TCF21 and Notch4 were greatly downregulated in lung cancer cells and TAMs, and Notch4 was the downstream target gene of TCF21.
Meanwhile, macrophages, as a type of innate immune cells, participated in the body's antiviral and antitumor effects. Under the context of tumor microenvironment, the macro-phages derived from peripheral blood are able to enter tumor tissues making it possible to form TAMs infiltrating tumor tissue. TAMs are an important component of tumor inflam-matory microenvironment, which can promote tumor invasion, metastasis and angiogene-sis[21]. TAMs can be polarized by cytokines and chemokines secreted by tumor cells and tumor stromal cells, allowing to form two phenotypes with opposite characteristics and functions. One is the M1 type with pro-inflammatory and anti-tumor activities[22]; The oth-er is the M2 type macrophages formed due to polarization which exert roles in an-ti-inflammation, tumorigenesis, growth, and metastasis[23]. Advanced lung cancer cells are infiltrated with a great deal of M2-polarized macrophages, and the TAM-targeting anti-body can substantially reduce lung tumor metastasis after acting on lung cancer mice[24,25]. The current research confirmed that the expression of TCF21, Notch4, Hes1, Hey1, TNF-α, IL-1β and p40 in M2 macrophages and TAMs was much lower than that in M1 macro-phages, whereas the expression of MRC1, FN1 and COX2 was remarkably higher than that of M1 macrophages.
Typical target genes of Notch signaling transduction include Hes and Hey[26]. It has been proven that Notch signaling transduction is related to anti-tumor immunotherapy[27]. TNF-α is a pro-inflammatory cytokine, which can be secreted by TAM and cancer cells. Meanwhile, TNF-α is also intimately correlated to the prognosis of NSCLC patients in the promotion of inflammation, epithelial-mesenchymal transition, invasion and metastasis under a tumor microenvironment. IL-1β is mainly synthesized and secreted by mono-cyte-macrophages, and plays an important role in body immune regulation[28]. Being a frequently applied tumor marker, P40 is usually expressed in the basal layer of epithelial tissues, and its expression is upregulated in squamous cell carcinoma tissues[29]. MRC1 is an M2-specific marker gene and highly expressed in M2 macrophages[30]. FN1 is expressed in stromal cells but mainly expressed in TAMs when in immune cells[31]. The overexpres-sion of COX-2 is related to tumorigenesis and inflammatory response[32]. The results of the present experiment revealed that M1- and M2-type polarization were successfully in-duced.
In recent years, the discovery of immune checkpoint blockers targeting lung tumors has effectively demonstrated the immunogenicity of NSCLC[33]. Although immune check-point blockers are widely used in the treatment of advanced NSCLC, most patients devel-op innate or acquired immune resistance during immunotherapy. Therefore, exploring new molecular targets that may reverse or stop immune escape will help further improve the prognosis and survival of patients with NSCLC. The immune memory generated dur-ing immunotherapy could achieve a long-term protection for the body, effectively prevent tumor recurrence and metastasis, which is expected to completely cure the tumor. The po-larization of macrophages exerts an important role in tumor immunotherapy. TAMs can support tumor development by interacting with T cells[34]. First, macrophages polarize to the M1 phenotype, making it possible to transform from a state of promoting tumor sup-pressive immunity to a state of anti-tumor immunity promotion, thereby creating a pro-inflammatory tumor microenvironment and improving the inhibitory effect of TME on immunotherapy[35]. M2-type TAMs produce immunosuppressive factors, thereby inhibit-ing the function of CD4+ and CD8+T cells. Meanwhile, it secretes chemokines to inhibit the anti-tumor response of the tumor immune microenvironment. The combination of between the regulation of macrophage phenotype and tumor immunotherapy can perfectly improve the efficiency of immunotherapy[36]. For instance, Kim et al. have enhanced the anti-tumor effect of the immune checkpoint inhibitor PD-L1 by polarizing M2-type macrophages in TAMs to M1-type[37], while Huang et al. applied the polarization of TAMs to enhance the treatment benefits of tumor vaccines[38]. This experiment demonstrated that the polariza-tion of TAM to M1 can promote the viability of CD8+T cells, while the polarization of TAM to M2 reduces the viability of CD8+T cells and tumor killing. Overexpression of TCF21 promotes the polarization of TAM to M1 and the increase of T cell viability by acting on the Notch signaling pathway. Certain studies have indicated a close relationship between the Notch signaling transduction and macrophage polarization[39]. Overall, our findings demonstrated that the downregulation of TCF21 in NSCLC might result in enhanced im-munosuppressive effect of tumor-associated macrophages, which is not conducive to tu-mor management.”
Comments on the Quality of English Language
Manuscript will benefit from English language editing services and requires a serious effort to add scientific details. In its current form, the language is far too lay for an audience of researchers and clinicians
Reply: Thank you very much for your careful review of this article, which will help us to further improve this study. We have refined the language of this article according to your suggestions.
Round 2
Reviewer 1 Report
Accept
Still require English editting
Reviewer 3 Report
The authors have addressed all the comments in detail in their revision. The revised manuscript provides additional context, details and clarity which have improved the story significantly. No further changes required.